# Binless normalization of Hi-C data provides significant interaction and difference detection independent of resolution

Yannick G. Spill[1,2], David Castillo[1,2], Enrique Vidal[1,2] & Marc A. Marti-Renom [1,2,3,4]

Chromosome conformation capture techniques, such as Hi-C, are fundamental in characterizing genome organization. These methods have revealed several genomic features, such as chromatin loops, whose disruption can have dramatic effects in gene regulation. Unfortunately, their detection is difficult; current methods require that the users choose the resolution of interaction maps based on dataset quality and sequencing depth. Here, we introduce Binless, a resolution-agnostic method that adapts to the quality and quantity of available data, to detect both interactions and differences. Binless relies on an alternate representation of Hi-C data, which leads to a more detailed classification of paired-end reads. Using a large-scale benchmark, we demonstrate that Binless is able to call interactions with higher reproducibility than other existing methods. Binless, which is freely available, can thus reliably be used to identify chromatin loops as well as for differential analysis of chromatin interaction maps.

[1] CNAG-CRG, Centre for Genomic Regulation (CRG), Barcelona Institute of Science and Technology (BIST), Baldiri i Reixac 4, 08028 Barcelona, Spain. [2] Centre for Genomic Regulation (CRG), Dr. Aiguader 88, 08003 Barcelona, Spain. [3] Universitat Pompeu Fabra (UPF), 08002 Barcelona, Spain. [4] ICREA, Pg. Lluís Companys 23, 08010 Barcelona, Spain. Correspondence and requests for materials should be addressed to Y.G.S. (email: yannick.spill@nsup.org) or to M.A.M.-R. (email: martirenom@cnag.crg.eu)

Since the invention of chromosome conformation capture (3C) experiments[1], our perception of the genome has become that of a structured but highly dynamic polymer[2]. In particular, Hi-C experiments[3] made it possible to quantify the frequency of contact between any two locations in the genome. We now know that the mammalian genome is organized into compartments which, in turn, are partitioned into topologically associated domains (TADs) that hold groups of genes. More recently, a series of Hi-C experiments with great sequencing depth revealed that, at the smallest scale, chromatin loops can form mainly between gene promoters and their enhancers or between CTCF bound loci[4]. Yet while it might at first seem that the detection of such events is a mere consequence of better experiments and increased sequencing efforts, the computational tools to detect them proved to be crucial. Indeed, the size, noise, and complexity of 3C-like experiments raised completely new research questions for statisticians and computer scientists. As a result, numerous methods have been developed to computationally analyze the results of 3C-like experiments[5].

Genome interaction matrices derived from Hi-C experiments[3] usually show strong systematic biases along both counter-diagonals and rows or columns. It is, therefore, customary to remove these biases through normalization procedures[6–15]. Two types of strategies exist to normalize Hi-C data, as was recently reviewed[13]. First, explicit methods assume that all biases affecting Hi-C data are known and can be provided as input to the normalization software; for example, HiCNorm[6] requires three genomic tracks for GC content, mappability, and fragment length. Second, implicit methods make the theoretical assumption of equal visibility for all loci[7]. They then deduce the biases that must be subtracted to recover normalized Hi-C matrices. Both approaches, however, depend not only the quality of the data but also on the quantity of sequencing reads that determines the genomic resolution to which interaction matrices will be normalized. This step is crucial, as genomic features such as TADs[16,17] or chromatin loops[4] are detected from normalized matrices. Unfortunately, there is no algorithm that is best for all analyses such as normalization, TAD or loop calling. A recent review[18] concluded that TAD detection is consistent across a broad range of algorithms but it differs mainly when TADs are nested, because different algorithms will choose different levels of nesting. Loop calling is, however, very inconsistent across methods, none of which stands out to be better than the others. Importantly, it was found that called interactions are poorly reproducible across technical or biological replicates. Overall, it is still best to perform redundant analyses with several methods to conclude the validity of a set of detected interactions.

To address these limitations, we introduce Binless, a method to normalize Hi-C data in a robust, resolution-independent and statistically significant way (see graphical overview in Fig. 1). Binless uses the negative binomial regression framework that proved valid in HiCNorm[6] and oneD[19] but estimates the genomic biases using only the input Hi-C data. To adapt to the size of the features present in the data, Binless uses the fused lasso algorithm originally implemented for image analysis. We show that the resulting normalized matrices by Binless, in addition to being visually simpler than regular Hi-C maps, allow for improved and reproducible interaction and difference detection.

## Results

**Binless rationale.** Detectable 3D genomic features have no specific resolution. For example, for mammalian genomes, compartments are of several megabases (Mb) in size (detected from matrices of ~100 kb resolution), TADs are of about 1 Mb in size (detected from matrices between 20 and 50 kb resolution), and chromatin loops are a few kb (detected from matrices of resolutions higher than 5 kb). In fact, algorithms to detect genomic compartments, TADs and loops are sensitive to the resolution of the input data[18]. Therefore, ideally the detection of any 3D genomic feature (including those yet to be discovered) needs to be done with bin-less interaction matrices, in which the data is fused in cells of varying resolution adapted to the features of interest. Binless aims to accomplish this by iteratively normalizing, smoothing, and fusing the data. The following sections describe the working principle of Binless.

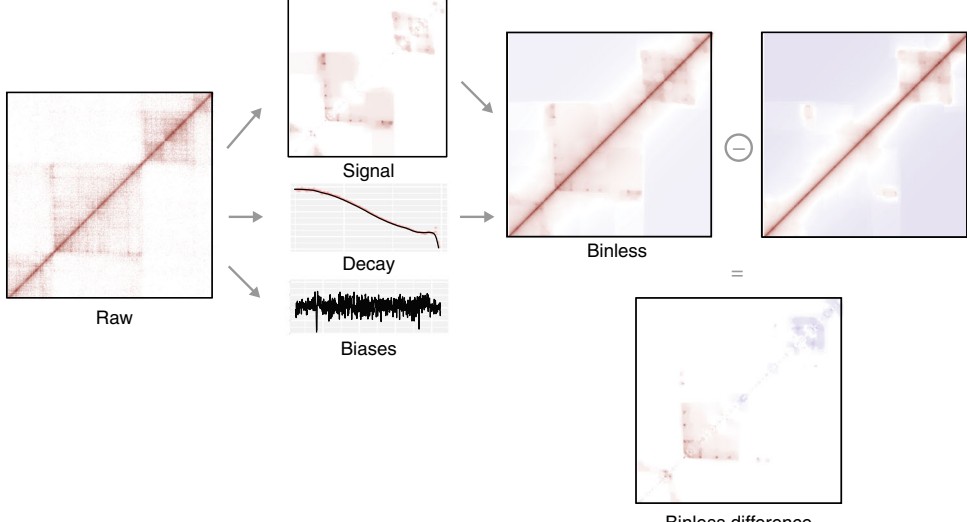

**Fig. 1** Main estimates computed by Binless. From the raw data, Binless estimates genomic and decay biases, as well as an additional signal (example on the TBX3 locus, see Sup. Fig. 9). This binless signal matrix reports contributions that are significantly different from what can be expected from genomic and decay biases alone. Its unit is a minimum fold change with respect to the background. Any thresholding operation should be performed on this matrix. The binless matrix is the combination of signal and decay estimates, and is useful for visualization. If another dataset was normalized simultaneously, a binless difference matrix can be computed. Similar to the binless signal matrix, its unit is a minimum fold change from one dataset to another. Only differences which cannot be imputed to noise alone are reported

**Working principle**. Prior to normalization, Binless estimates a series of biases which correspond to the background model against which the raw data will be normalized. Three of such biases form the core of the procedure (Fig. 1 and Supplementary Fig. 1). First, genomic biases are estimated to model the varying coverage of the experiment across the genome and are modeled as a smooth function that depends on the genomic position. Second, the diagonal decay is estimated to capture the decrease in average interaction frequency as loci separate in sequence, which smoothly decreases as the distance between the interacting loci increases. And third, the residual signal is estimated to detect local features, such as TADs and loops. It is important to note that the signal is also resolution-independent. To estimate it, Binless corrected data by the genomic and decay biases is collected at a very high resolution, which needs to be smaller than the smallest detectable feature. Next, the fused lasso algorithm[20] fuses neighboring pixels if they have a similar signal. The fused lasso algorithm is an alternative approach to other neighborhood filtering approaches[21,22], for which highly efficient implementations are available[23]. The resulting signal matrix (Fig. 1 and Supplementary Fig. 2A) is a collection of patches of varying sizes and shapes.

Next, to correct the input interaction matrix, Binless uses an in iterative correction similar to ICE[7], but with no assumption of equal visibility for all loci. In fact, it uses a negative binomial count regression framework, similar to HiCNorm[6] or oneD[19], which allows row and column sums of a Hi-C matrix to deviate from a reference value. Note that Binless models the accumulation of all local genomic biases in a non-parametric way by not regressing against external data such as GC content or mappability but by building on a popular class of regression models, called Generalized Additive Models[24,25]. Binless uses a negative binomial likelihood, a common choice for Hi-C[6,19,26], as confirmed by recent experiments on Syn-HiC[27]. The genomic and decay biases are estimated using p-splines[28], whose smoothness adjusts to the quantity of data, and therefore is less prone to over- or under-fitting. The use of smoothing splines is justified when normalizing sparse Hi-C datasets, especially if 4 letter cutters are used since the number of possible contacts is so large that even very dense datasets, such as the kilobase-resolution datasets of Rao et al.[4], only accumulate about 1 contact every 10 cut site intersections (Supplementary Fig. 3). To ensure proper normalization and to avoid overfitting, it is therefore essential to share information spatially, which is what Generalized Additive Models were designed for. To show this feature in the Hi-C context, we took different sub-samplings of the SELP locus. Generalized Additive Model ensured that biases stay as smooth as possible (Supplementary Fig. 4).

Binless is also robust to sequencing depth as it does not over fit. To test these features, we normalized the human chromosome 22 using various amounts of data, ranging from 1 to 100% of the combination of 7 IMR90 replicates of Rao et al.[4] (Supplementary Fig. 5). When more data is added, features start to be visible in the raw data. Binless retains these features only when it can be excluded that they are caused by noise fluctuations. Then, TADs and loops are detected simultaneously. At no moment does Binless follow all the fluctuations in the data, because the statistical formulation uses Generalized Additive Models[20,24], which prevents it. The same observations hold for the genomic and decay biases (Supplementary Fig. 4).

**Benchmark**. Do binless matrices result in more reproducible Hi-C analysis? In line with a recent analysis of several Hi-C normalization methods[18], we analyzed 41 different Hi-C datasets of varying sequencing depths, restriction enzymes, cell types and organisms (Supplementary Data 1). We compared Binless to other methods by computing several metrics on selected pairs of datasets (Methods). The stratified correlation coefficient (SCC)[29] was highest with Binless, and remained high even at 5 kb resolution when comparing biological replicates (Fig. 2). Methods that do not rely on smoothing, such as ICE[7] or oneD[19], were able to better reproduce datasets at 100 kb resolution, compared to raw data. However, reproducibility degraded for matrices at higher resolutions. In contrast, methods relying on the fused lasso (Binless and HiCRep[29] lasso modification[30] used in HiC-bench[31], hereafter named HiCRep) showed a marked improvement at all resolutions. For Binless, the median SCC was larger than 0.98 at all resolutions. Reproducibility was also high across restriction enzymes (Supplementary Fig. 6B) with a median SCC larger than 0.97 at all resolutions. Other metrics and comparison types showed similar trends (Supplementary Fig. 6C–K), suggesting that Binless increased the reproducibility of Hi-C analysis.

Do binless matrices result in improved interaction detection from Hi-C matrices? Using the benchmark described above, we next examined the number of true positives detected by Binless and other methods (Fig. 3 and Methods). At 5 kb resolution, Binless recalled 10% of all annotated true positives on average. The second-best method only recalled 0.8% on average. This significant improvement in sensitivity was achieved while maintaining the false positive rate below 2.5% on average (Supplementary Fig. 7 and Supplementary Fig. 8 for side-by-side examples). The results, thus, indicate that Binless achieved high specificity in our benchmark.

Can binless matrices be used to detect differences between two Hi-C experiments? Using the just described benchmark, we next computed the sum of all significant differences between either technical replicates, or different cell type experiments (Fig. 4). Binless detected higher number of differences between experiments from different cell types than between technical replicates, even at high resolution (Wilcoxon one-sided $p < 10^{-14}$). The resulting differential matrices provide a clear and quantitative representation of changes between two experiments (Fig. 5 and Supplementary Fig. 9).

**Alternate representation and classification**. The origin of Binless stemmed from representing Hi-C data at very high resolution, which resulted in interesting patterns. For example, the Hi-C map of the *Caulobacter crescentus* genome[32] at 100 base-pair resolution shows highly dense square patterns at the junction of two restriction sites (Fig. 6c). These patterns prompted us to introduce an alternate representation of Hi-C data (Fig. 6d). In this representation, each read was displayed as an arrow in the 2D plane. Projecting the arrow onto the diagonal along the $x$ or $y$ axis, we could retrieve the start, end and orientation of each of the two mapped read pairs in an interaction (Fig. 6b). Contrary to representing Hi-C data as a matrix of read counts at a given resolution, this base-resolution representation gave insight into the way paired-end reads align around each cut site. This also prompted us to classify each of the interactions (or arrows in the alternate representation) into two large categories, according to whether they gather in the immediate vicinity of the diagonal or not (Fig. 6d). First, arrows that were far from the diagonal correspond to read pairs with successful re-ligation (or, rarely, mapping errors). They could be further subdivided into four contact categories: "Up" contacts, which are upstream of the cut-site intersection; "Down" contacts, which are downstream of the cut-site intersection; "Close" contacts, which are closer from the diagonal than the cut-site intersection; and "Far" contacts, which are further from the diagonal than the cut-site intersection. Second, arrows that clustered close to the diagonal corresponded to

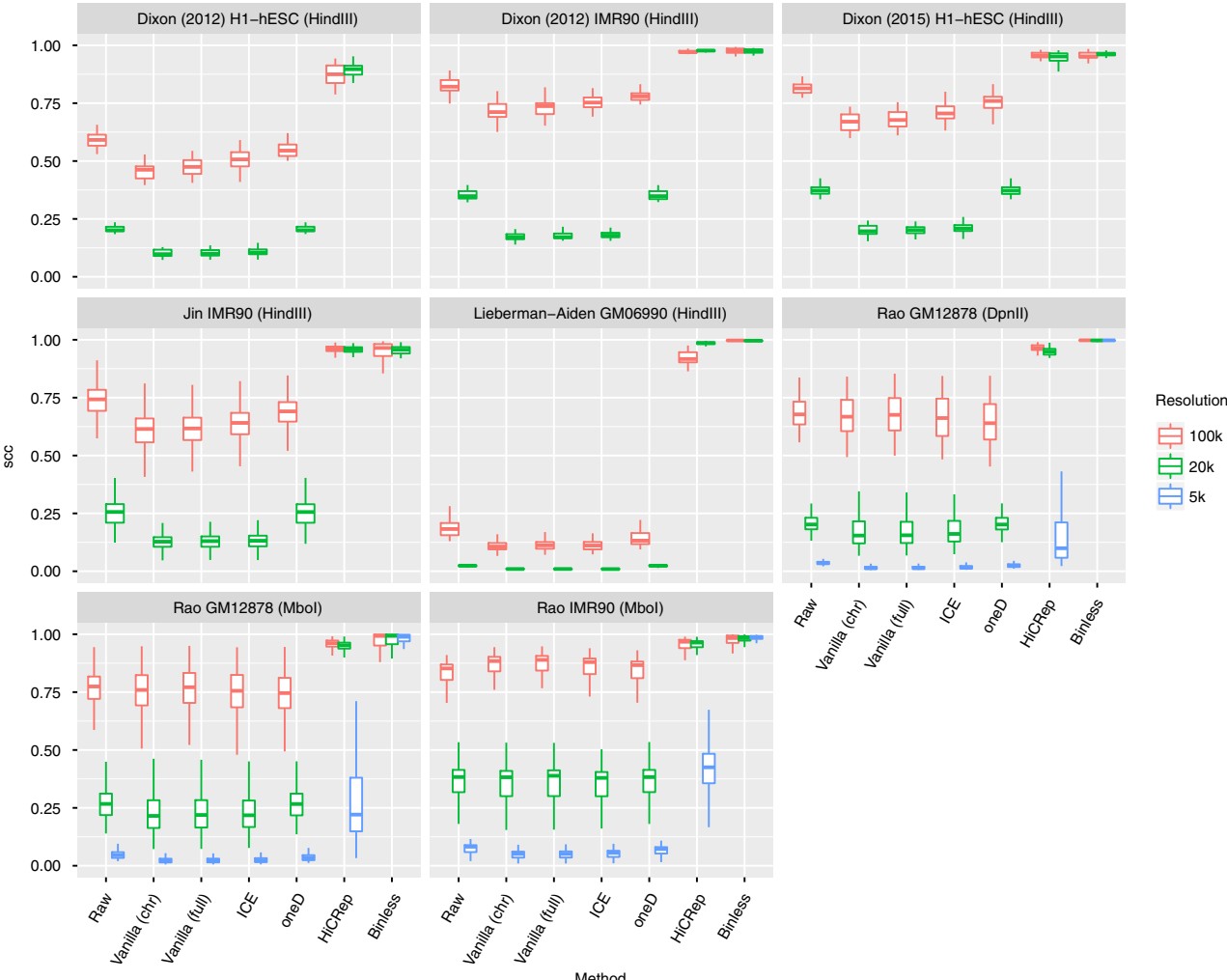

**Fig. 2** Binless matrices show high correlation across biological replicates. Here we report the stratified correlation coefficient (SCC) between two biological replicates. The SCC was computed on each chromosome, for each resolution, normalization method, and dataset. Boxplots show the distribution of SCC values across a large number of comparisons (see methods for details, Sup. Table panel 3 for comparisons and panel 8 for sample sizes)

read pairs in which ligation events were unsuccessful, or which resulted in the re-ligation of the same piece of DNA that was just cut. Depending on their position and orientation relative to a nearby cut site, a classification was proposed (Fig. 6a and Supplementary Fig. 10). For example, the so-called dangling reads (that is, reads containing fragments of DNA that were digested but not re-ligated) were arrows that stack along the coordinates of a cut site. This classification allowed computing two key Hi-C quality diagnostics that serve as input to the next steps in Binless. First, the distribution of sonication fragment lengths was gathered from reads close to the diagonal (Supplementary Fig. 11A), which were used to detect problems during the sonication step of the Hi-C protocol. Second, the precise starting points of the dangling ends was also gathered (Supplementary Fig. 11B), as they are specific of each restriction enzyme. Spurious peaks in these plots could be indicative of DNA degradation, or problems during data processing. Additionally, this representation allowed also to detect contacts between sites closer than 1 kb in sequence, which cannot be modeled by Binless, and as such can be removed beforehand (Supplementary Fig. 11C).

Finally, it is important to note that this alternate representation also allowed us to assess some of the biases to be removed during the normalization procedure. For example, in a Hi-C experiment, it is expected that the number of dangling reads drops with

increased efficiency of ligation at a particular cut site. The proportion of the different types of dangling reads correlates with such biases and, as such, can be used during normalization (Fig. 6d). In fact, Binless counts the number of reads in each dangling category at each cut site intersection, which are later used as input to the normalization procedure. In other words, the number of dangling reads is used to compute the genomic biases at cut-site level.

## Discussion

A number of problems arise in binned interaction detection as the significance of interactions depends on the chosen resolution. In fact, loops are usually called at 1–10 kb, TADs at 50–100 kb and compartments at 100–1000 kb resolution[18]. Unfortunately, the best resolution at which to call a particular genome structural feature is still an open question, and may also depend on data quantity/quality. Importantly, at typical sequencing depth for Hi-C experiments, the number of common called interactions between replicates is low[18]. To address these limitations, the resolution of a Hi-C matrix can be chosen based on the distance between two loci of interest[5]. Indeed, it is expected that higher resolution can be reached close to the matrix diagonal, because sequencing depth is what dictates where to fix the tradeoff

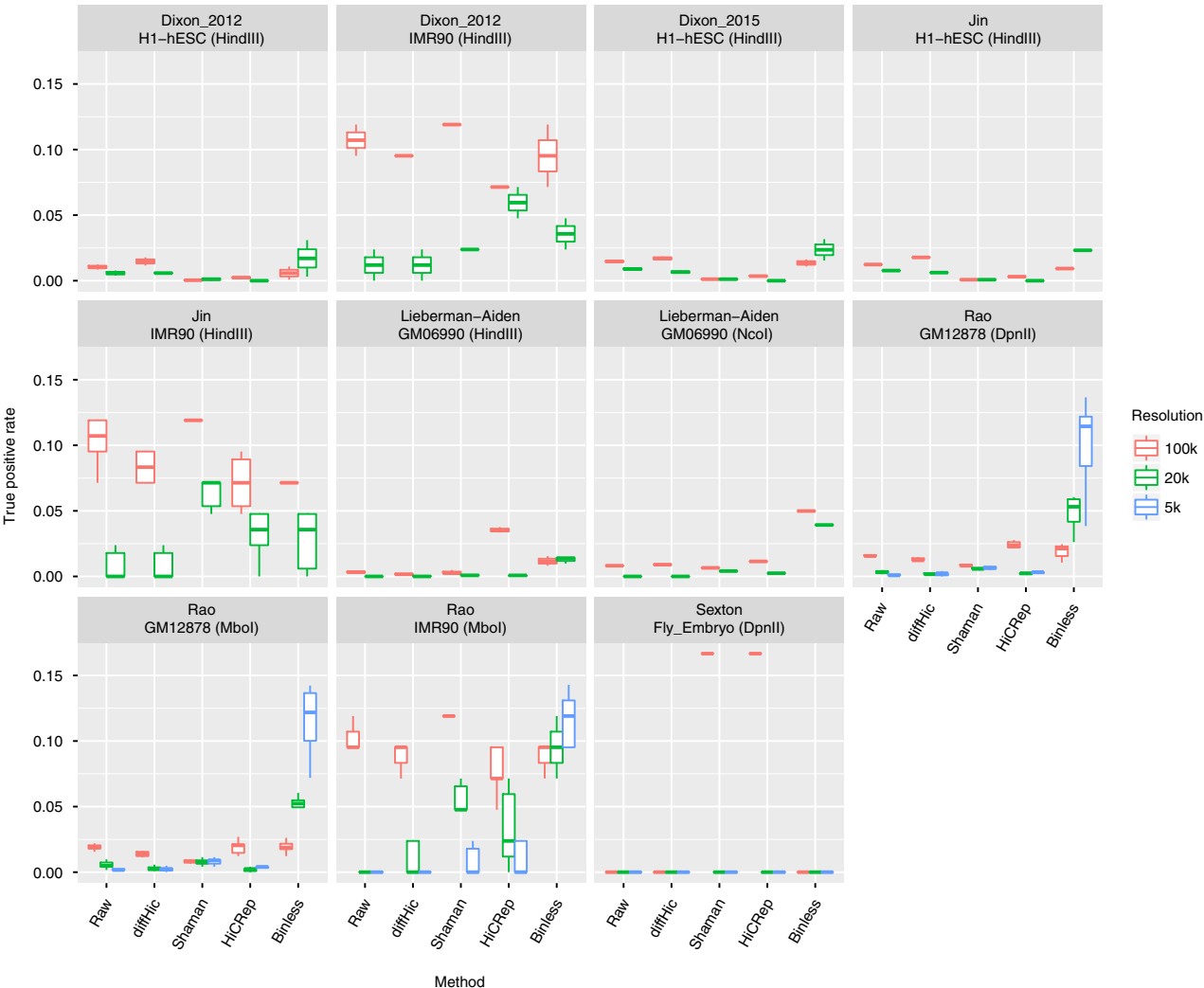

**Fig. 3** Binless interaction detection has an improved true positive rate. Each box reports the true positive rate at several resolutions, for each interaction detection method. (see methods for details, and Sup. Table panel 9 for total number of annotated true positives)

between high resolution and genomic distance. With Binless, it is now possible to perform normalization, interaction, and difference detection entirely without specifying a Hi-C matrix resolution. Internally, Binless adapts the "resolution" of the detected features depending on their position in the Hi-C matrix (Supplementary Fig. 2A), which avoids the tradeoff between resolution and genomic distance. In fact, the fused lasso algorithm used for that purpose ensures that, at each position, the local bin size is neither too big, which could lead to averaging out some features, nor too small, which would increase the noise. For example, Binless is able to highlight both loops and TADs within the same binless matrix (Supplementary Fig. 8).

Here, we prove that it is possible and advantageous to normalize Hi-C data in a resolution-agnostic way, using binless matrices. However, how can the quality of a dataset be assessed? Binless matrices have a base resolution, which can be seen like the pixel size of a detector. These pixels are then fused when their signal contributions are similar. Contrary to HiCRep, we employ a weighted version of the fused lasso algorithm. This choice is important, because it allows the fusion effect to be weak where most of the reads accumulate, but to be strong where no data is present. The size of patches formed by the fused lasso algorithm therefore varies substantially (Supplementary Fig. 2). Close to the main diagonal, where most of the pair-wise interactions map, the

matrix is enriched in small patches (or higher-resolution features such as loops). Far away, the data is scarce and patches become larger (or lower-resolution for TADs and compartmentalization). Thus, the effective resolution of binless signal matrices depends on the distance from the diagonal, and therefore adapts to the quantity and quality of data. In fact, the resulting patches have an approximately constant read density, independent of patch size (Supplementary Fig. 2D). We therefore propose to use this average read density per patch as a proxy for dataset quality.

Binless signal matrices result after suppression of diagonal decay and the compensation for genomic biases in a Hi-C raw interaction matrix. To accomplish this, Binless performs two main steps (Methods). First, an un-thresholded signal matrix is estimated (in logarithmic scale) along with its fusion strength parameter, $\lambda_2$. And second, the algorithm estimates a significance threshold, $\lambda_1$, which is used to obtain the final signal matrix by a so-called soft-thresholding operation. In this case, soft-thresholding corresponds to setting to zero all regions whose log-signal is lower than $\lambda_1$, and subtracting $\lambda_1$ from the remaining values. Therefore, when there is not enough evidence for signal in a given region, the binless signal matrix will be zero (Supplementary Fig. 5). When evidence is strong enough, the reported signal represents by how much, at minimum, local contacts are enriched with respect to what would be expected by local genomic

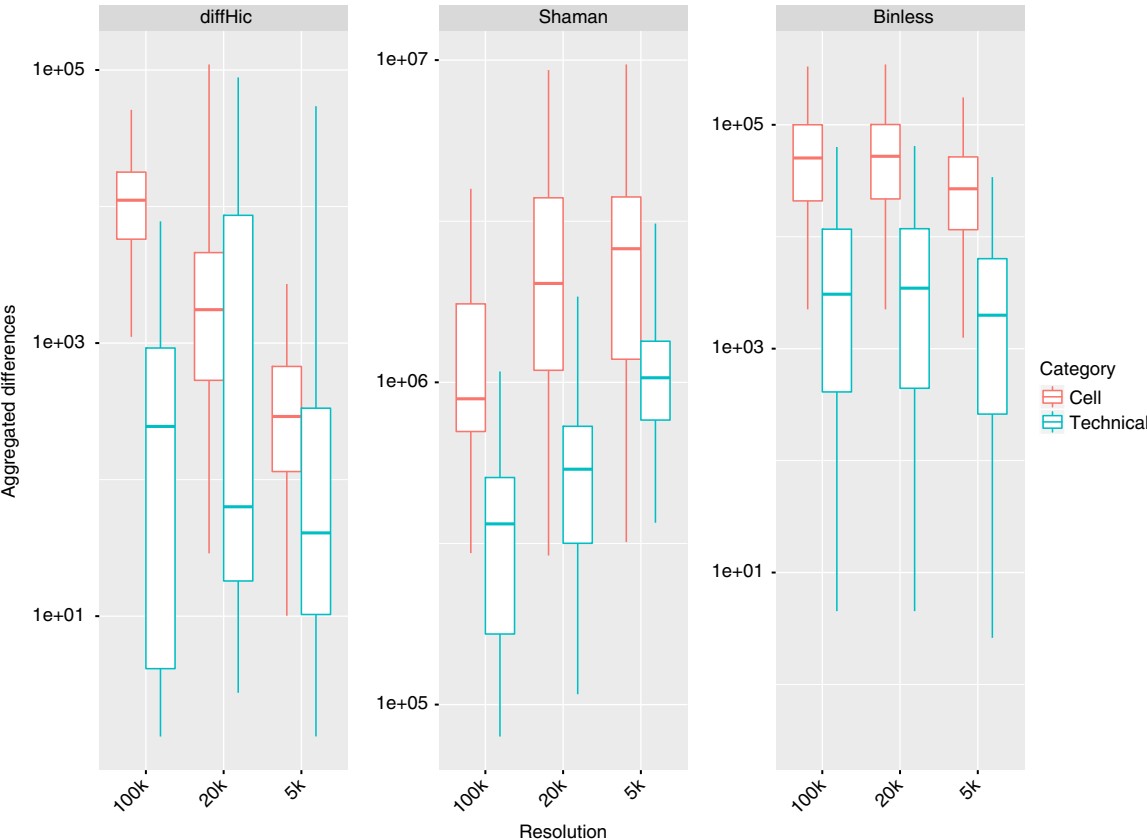

**Fig. 4** Binless differences are more numerous between cells than between technical replicates. See methods for details and Sup. Table panel 10 for total number of difference calculations

biases and the average interaction frequency at that distance. Deciding on what is noise and what is signal is the role of the Generalized Additive Model, and is reflected by the value of the $\lambda_2$ parameter in signal detection (and similar parameters in the genomic and decay biases). As shown also in HiCRep[33], when $\lambda_2$ is large, fusion is strong, and patches become large, even close to the diagonal. When $\lambda_2$ is small, fusion is weak and the matrix becomes less smooth and closer to the raw data. Binless spends a large amount of time to determine this parameter, employing exact solutions for the biases, and the Bayesian information criterion (BIC) for the signal and difference estimates. These criteria consider the need to fit the data on one hand, and the need for smoothness on another hand. The final value of $\lambda_2$ will depend on both the quantity of data, and the estimated variability it contains. We should note that Binless does not "pick" loops. Since Binless is meant to be a locus-specific method, manual inspection is still required. If loop detection is required, the binless signal or difference matrix can be used to define loops at a given user defined threshold.

Here, we have introduced a statistically sound method to compute normalized binless interaction matrices derived from Hi-C raw datasets. The method stems from an alternate representation of Hi-C datasets, which in turn results in a modified classification of interactions between loci in a genome. Binless has been implemented in a R package and can be used in computational settings with high memory at the chromosome level. We have shown that this method is able to increase the reproducibility of Hi-C experiments and is able to more reliably detect statistically significant interactions in real-scenario experiments. Binless can be used to detect several structural features in the genome ranging from few kilobases (i.e., loops between two loci) to megabases (i.e., TADs or compartments). Finally, using the same statistical approach, Binless is able to detect differential interactions between two or more experimental datasets. Overall, we trust Binless is complementary to existing normalization methods for 3C-based experiments.

## Methods

**Base-resolution view of Hi-C data**. Paired-end reads are processed using the TADbit pipeline[14]. The input to Binless is the reads intersection file, which contains the genomic location, length, and strand for both ends of each read, as well as the coordinates of the closest upstream and downstream cut sites. It is assumed that the first read is always upstream of the second read. Duplicate reads are removed when reading the inputs (Supplementary Fig. 1a). At this step, the user should provide sonication fragment length and dangling end positions, which are also obtained by Binless from the reads intersection input file. These reads are then classified as shown schematically in Fig. 6 (see also Supplementary Fig. 10 for a decision tree). We define several categories. A left or right "dangling read" is a DNA molecule that starts or ends, respectively, on a cut-site, with both ends mapping on opposite strands. A "rejoined read", which has likely been re-ligated, spans across a restriction site. A "self-circle" corresponds to ligation of the two ends of a fragment. "Random reads" align close to the diagonal and on the same fragment, and point towards the diagonal in the base-resolution representation. They are thought to be genomic DNA, and are not specific to the Hi-C experiment. Most importantly, there are four "contact types", depending on which quadrant of the intersection between two restriction site they can be found in. "Up" and "Down" contacts are such that both read ends align upstream and downstream, respectively, of the closest restriction fragment. "Close" and "Far" contacts are closer or further, respectively, from the diagonal than the intersection of their cut sites. All contact types must point towards the restriction intersection in the base-resolution representation. Note that for neighboring cut sites, self-circles replace the close contact category. Finally, reads that cannot be classified (because they are too far from a restriction site, or because their direction does not match) are put in the "other" category.

**Exact model**. The negative binomial regression we employ has likelihoods of the following form (see Supplementary Methods for a complete overview)

$$d_i \sim NB(\mu_i, \alpha) \tag{1}$$

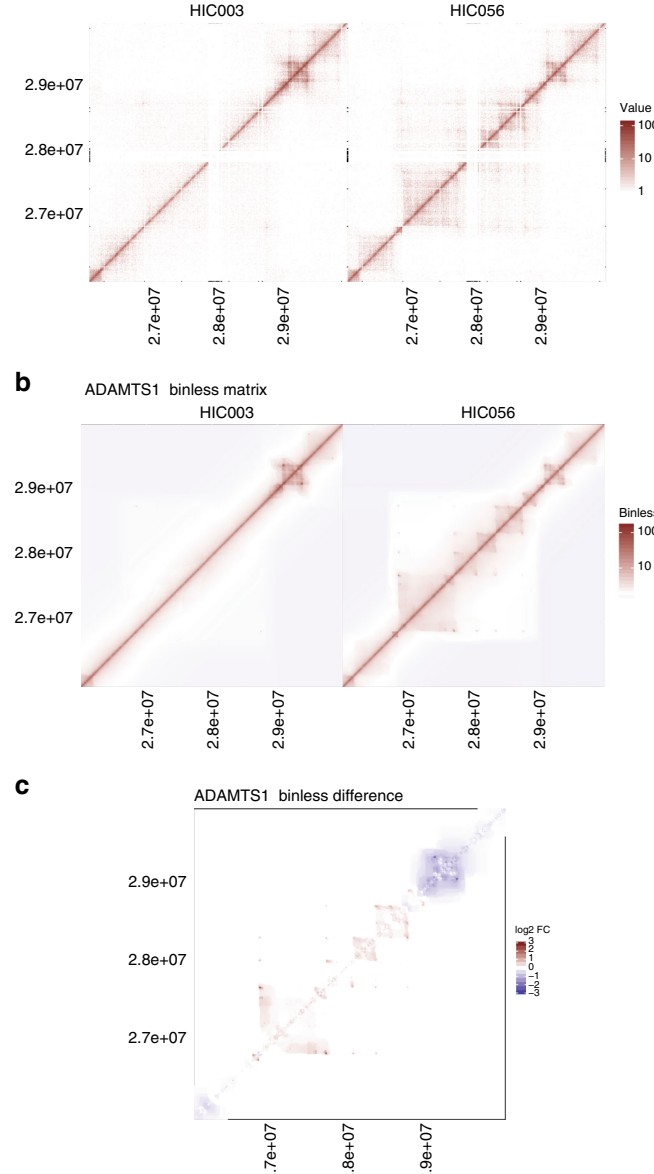

**Fig. 5** Example of difference calculations with Binless. ADAMTS1 locus (chr21:26M-30M, Rao HIC003 vs HIC056). **a**: Raw data. **b**: binless matrix. **c**: binless difference matrix

$$c_{ij} \sim NB\left(\mu_{ij}, \alpha\right) \qquad (2)$$

where $d_i$ is the number of dangling or rejoined reads at cut site $i$ and $c_{ij}$ is the number of reads in one of the four contact categories, observed between cut sites $i$ and $j$. $\mu_i$ and $\mu_{ij}$ are the respective means, to be estimated, and $\alpha$ the dispersion parameter of the negative binomial. The means $\mu_{ij}$ are parametrized using three "background" splines $\iota$, $\rho$, and $f$, and one "signal" term $s$, as we now explain. The efficiency of detection of a particular contact has been shown to be decomposable into genome-specific biases for each of the two reads in a read pair[7]. For reads aligning to the left (respectively right) of a cut site $i$, the number of contacts involving $i$ are therefore made proportional to their genomic bias $\iota_i$ (respectively $\rho_i$). $\iota$ and $\rho$ are modeled using p-splines[24,28,34]. The polymer nature of chromatin is thought to make Hi-C contact probabilities decrease with the genomic distance between two cut sites. Therefore, the number of contacts involving cut sites $i$ and $j$ are made proportional to the decay bias $f_{ij}$, which is forced to decrease with the genomic distance between $i$ and $j$. We use a smooth constrained additive model[35] for $f$. When the ligation efficiency for a cut-site decreases, one can expect a depletion in the number of contacts and an enrichment of dangling ends. Therefore, dangling ends are made to follow the opposite trend of the counts, and are

biased by $\iota$ for left-dangling and $\rho$ by right-dangling ends. Rejoined ends follow the (geometric) average bias at this cut site. Finally, a sparse 2D term $s_{ij}$ is meant to fit the signal that departs significantly from the background modeled by the genomic and decay biases. This term is modeled using the sparse 2D generalized fused lasso on a triangle grid graph[36].

**Optimized Binless**. Ideally, all parameters are optimized together. However only small datasets (less than about 100 cut sites) can be normalized in this way. For even the smallest Hi-C loci, it is necessary to model the contribution of cut-site intersections with zero observed counts implicitly. We combine this implicit representation with a fast coordinate descent algorithm. We refer to this implementation as "optimized Binless". In a nutshell, instead of optimizing all parameters at once, we optimize parameters relevant to genomic biases, diagonal decay, dispersion and signal (using gfl[37]) separately and iteratively. In each separate optimization, we compute the biases not using the individual counts, but using weighted average log-counts. This grouping by rows, counter diagonals or signal bins is what allows the computation to be orders of magnitude faster. Grouping is made possible by a repeated normal approximation to the log likelihood of the counts. This method, known as Iteratively Re-weighted Least Squares (IRLS) is very common in all types of generalized regressions[38]. Note that IRLS converges to the maximum posterior estimate. Therefore, the only approximation in this model is the implicit representation of zeros, which is similar to a mean field approximation for the Ising model.

The estimation of the dispersion is done differently. For a number of matrix rows (default 100), the maximum likelihood estimate of the dispersion is estimated on all counts (including zeros), dangling and rejoined reads according to the exact model. The final dispersion estimate is their median. In optimized Binless, the dispersion, biases, decay and corresponding stiffness penalties are optimized first, holding the signal fixed to zero. Upon convergence, the dispersion, biases and signal are then fitted, with a fixed fusion penalty ($\lambda_2 = 2.5$ by default) for the signal.

Upon convergence, two options are provided. If one seeks to obtain binless signal matrices, they can be estimated along with their fusion ($\lambda_2$) and threshold penalty ($\lambda_1$). If one seeks differences with respect to a reference matrix, or a group of matrices (e.g. grouped by condition), an extended model is proposed to compute it (Supplementary Materials). In this model, all matrices (or groups) have the same mean than the reference up to a difference term. Fused lasso is then applied on this difference term. By incorporating the difference within the probabilistic framework, we are able to maintain an accurate weighting and control the contributions of datasets relative to each other. This step is key to obtain difference matrices that can be interpreted in terms of "fold change", like the signal matrices.

**Fast Binless**. Optimized Binless is suited only for individual loci (0–3 Mb for 4-cutters) in which high precision is required. For chromosome-wide analyses as presented here, a tradeoff is proposed as follows. Data is binned at the chosen base resolution, and an IRLS scheme estimates the diagonal decay and biases along each binned row until convergence. To speed up the calculation and lower the memory footprint, an option is provided to limit the normalization to a certain interaction distance. Then, the signal and biases are estimated until convergence. The dispersion ($\alpha$), fusion ($\lambda_2$), and threshold penalty ($\lambda_1$) must be supplied to the call. Fast binless makes it possible to normalize whole chromosomes at 5 kb base resolution in a few hours (Supplementary Fig. 12).

**Estimation of parameters for fast binless normalization**. A procedure is provided to generate sensible values for the dispersion ($\alpha$), fusion ($\lambda_2$), and threshold penalty ($\lambda_1$) parameters (see supplementary methods). In a nutshell, several loci are selected from the chromosome to be normalized. For signal detection, this selection is based on the standard deviation of their directionality index (DI)[16] (Supplementary Fig. 13); for difference detection, it is based on a fast binless estimate of the difference computed at a fixed value of $\lambda_2$. Selected loci are subsequently normalized independently with optimized Binless. These normalizations are used to propose a set of parameters that will produce a similar binless signal or difference matrix with fast Binless (Supplementary Fig. 14).

**Available outputs**. Once several datasets have been normalized together, a number of matrices can be produced at any resolution (Supplementary Fig. 15). Decay and genomic bias matrices correspond to the estimated background terms, averaged over bins at the specified resolution. The normalized matrix corresponds to correcting the observed data by all genomic biases. It comes with corresponding error estimates, which are provided using the IRLS approximation (Supplementary Methods).

Binless signal matrices are the signal term obtained during normalization (Fig. 1). They can also be recomputed at a different base resolution afterwards. Their unit is a minimum fold change with respect to the background. Because sparsity was enforced while estimating the signal, the resulting matrix is nonzero when the signal is statistically significant. Should a more stringent significance threshold be applied afterwards, it must be applied on the binless signal matrix.

The binless signal matrix can be shown with an added decay bias. Such a matrix, which we simply call binless matrix, is visually closer to the raw data (Fig. 1 and Supplementary Fig. 9), but its unit is a fold change with respect to a

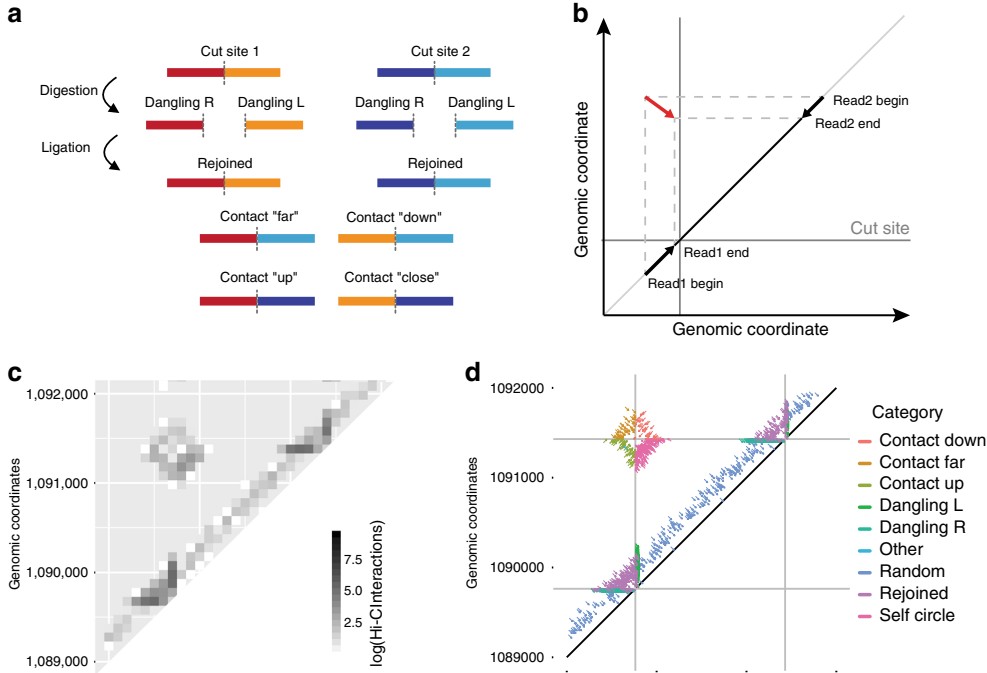

**Fig. 6** Classification of reads into categories used for Binless. **a** Reads are classified into dangling, rejoined ends and contacts. **b** Each paired-end read is represented by an arrow. Horizontal and vertical projections on the diagonal reveal the position and direction in which the read was mapped. **c** Zoomed Hi-C map of *Caulobacter crescentus* at 100 bp resolution with the apparent cut-site enrichment of interactions as a square in in the matrix. **d** Same data as in **c** now represented using arrows, with colors according to the Binless classification. Vertical and horizontal lines are cut site locations, while diagonal gray line represents the diagonal of the Hi-C matrix

background without diagonal decay. Finally, binless differences between datasets can be computed (Fig. 5 and Supplementary Fig. 9), and their unit is a minimum fold change between two datasets. All aforementioned matrices can be grouped (e.g. by condition) to improve the detection sensitivity.

**Recommendations**. In designing Binless, we attempted to minimize the number of free parameters. Yet, some of them are left to the choice of the user. As a general rule, their choice should not impact the resulting normalization. For example, the number of iterations should be large enough to reach convergence of the algorithm, which can be monitored using diagnostic plots Binless provides. Most importantly, the number of basis functions per kilobase controls the maximum wiggliness of the genomic biases. If it is large, the computational burden is high and the normalization can, for very large datasets, become unstable. If it is too small, the genomic biases will not be estimated properly. We suggest to start with a value of 50. Similarly, for binless detection, the base resolution should be as small as the smallest feature one hopes to detect. Out of computational considerations, we recommend a base resolution of 5 kb for 4-cutters, and 20 kb for 6-cutters or low-coverage 4-cutters. It is important to keep in mind that the base resolution gives the size of the smallest feature one hopes to detect. Lowering it might be attractive at first, but optimization is 4 times more difficult every time the base resolution is divided by 2. Once normalized, the data can be re-binned if necessary.

**Data processing**. We processed all datasets presented in this paper using the TADbit pipeline[14] and Binless 0.13.0 (Supplementary Data 1, first two panels). Whole-chromosome normalizations and differences were performed by first determining the proper parameters on submatrices along the diagonal, and then using fast Binless with these parameters on the whole chromosome (see above). Binless matrices were obtained at their nominal base resolution, and if necessary re-binned at a lower resolution. For subsampling of the data in Supplementary Figs. 4 and 5, we took a subset of all available reads by drawing the read count from a binomial distribution (coin tossing). Each dataset was then normalized independently.

Raw matrices corresponded to reporting the number of observed reads per bin (5, 20, or 100 kb resolution) after filtering with TADBit. ICE matrices corresponded to applying the iterative correction algorithm[7] on genome-wide raw matrices at the specified resolution. Vanilla matrices were obtained after the first iteration of ICE, either on a whole-genome matrix (vanilla full) or a matrix per chromosome (vanilla chr). OneD matrices were computed according to the algorithm of Vidal et al.[19] oneD, ICE, and vanilla matrices were computed using the dryhic 0.0.0.9000 R package[19]. HiCRep and HiCRep z-score matrices[30] (e.g. distance-normalized) were computed using the efficient high-resolution implementation based on gfl[37], kindly provided by the authors of HiC-bench and by following the

optimization method suggested in the paper[30], with slight modifications. For each chromosome, the ICE-corrected matrix was used as input, and the algorithm applied with 11 different values of the smoothing penalty $\lambda$. At 5 kb resolution, 11 values were chosen equally-spaced between 0 and 1. At 20 kb, they were chosen between 0 and 10. At 100 kb, between 0 and 100. Then, the stratified correlation coefficient (SCC)[29] was computed on matrices with successive values of $\lambda$. A one-tailed Wilcoxon test was computed on the SCC values of all chromosomes for a given pair of successive $\lambda$ values. The optimal $\lambda$ is the largest one for which the $p$-value is <0.001. DiffHic enrichment matrices[26] were obtained as the raw chromosome-wide matrices converted to ContactMatrix format, using a count filter of 1 and not storing neither zeros nor NAs. Enriched pairs were called using a flank width of 3. The R package diffhic 1.10.0 was used. Difference matrices are obtained as follows: Raw chromosome-wide matrices were converted to ContactMatrix format as described above, and the two datasets merged together. Non-linear normalization using LOESS was performed. In absence of replication, we performed a simple GLM fit with a dispersion of 0.01, followed by a likelihood ratio test. The difference matrix reported the minus log10 Benjamini-Hochberg-adjusted $p$-value. Shaman score matrices[15] were converted from mapped and de-duplicated reads obtained by TADbit to Shaman input (tab-separated chromosome, start, end for read 1, same for read 2, and an extra undocumented column of ones). Individual datasets were then shuffled and scored using default options. The R packages Shaman 2.0 and misha 4.0.2 were used. Shaman difference matrices were computed by subtracting the score matrices.

**Benchmark: comparisons**. We normalized all 41 Hi-C datasets presented in a recent Hi-C benchmark[18] (Supplementary Data 1) with several different tools, including Binless. We subjected all datasets to pairwise comparisons, by chromosome, for a number of normalization methods. Reproducibility was assessed using one of four metrics, as done in ref. [19]. First, the stratum-adjusted correlation coefficient (SCC)[29] was computed with a distance cutoff of 5 Mb (as in the original paper). Second, the reproducibility index[39] was computed on the 15 first components. Third, the Pearson correlation was computed between matrices whose value at $(i,j)$ is the original value divided by the average of all values at the same genomic distance than $(i,j)$, with a distance cutoff of 5 Mb. Fourth, the Spearman correlation was computed between matrices with a distance cutoff of 5 Mb.

Three classes of pairwise comparisons were formed between datasets (Supplementary Data 1, panels 3–5): biological replicates, technical replicates, and same cell type but different enzyme. Matrices subject to these comparisons all contain a strong diagonal, and are not distance-normalized. The methods compared were: raw data, one iteration of ICE (i.e., vanilla) applied to a chromosome, vanilla on a whole genome, ICE on a whole genome, oneD on a whole genome, HiCRep by chromosome, and Binless by chromosome. For Binless,

normalization was performed at 5 kb or 20 kb base resolution, and matrices re-binned to lower resolutions (20 kb and 100 kb). Other matrices were produced by directly performing the corresponding normalizations at 5 kb, 20 kb, and 100 kb resolution. Results are shown in Fig. 2, Supplementary Figs. 6 and 8. Sample sizes are reported in Supplementary Data panel 8. When shown, boxplots report the median (center line), first and third quartile (lower resp. upper hinges) and largest (smallest) value no further than $1.5 \times$ IQR (interquartile range) from the hinge (upper resp. lower whisker).

**Benchmark: interaction detection.** A list of more than 2800 true positive or true negative interactions obtained by 3C, 5C, ChIA-PET, and FISH was compiled in a recent benchmark[18] and was kindly provided by the authors (Supplementary Data panel 9 reports the number of annotated interactions). The true positive (resp. true negative) rate was computed by intersecting available true positive (resp. true negative) interactions in that cell type with the top 0.1% of interactions in a given matrix. The methods compared were: raw data, diffHic enrichment, Shaman score HiCRep z-score and Binless signal matrices. All these matrices, except the raw data, are distance-normalized. As previously, resolutions were 5 kb for 4-cutter datasets (Supplementary Data 1), 20 kb and 100 kb for all. Results are shown in Fig. 3, Supplementary Figs. 7 and 8.

**Benchmark: difference detection.** Pairs of datasets were tested for significant differences. Two groups of datasets were formed (Supplementary Data 1, panels 6 and 7): comparisons between technical replicates, and comparisons between different cell types. We compared diffHic, Shaman and Binless by reporting the sum of all difference scores on each matrix. For diffHic difference matrices, we use all the minus log10 Benjamini Hochberg $p$-values if they satisfy $p < 0.05$. For Shaman difference matrices, we use all absolute differences which are larger than 30. For Binless significant difference matrices, we use all nonzero absolute log10 differences. Results are shown in Fig. 4 and Supplementary Fig. 9. Total number of difference computations reported in Supplementary Data 1, panel 10.

**Reporting Summary.** Further information on research design is available in the Nature Research Reporting Summary linked to this article.

## Data availability
All relevant data supporting the key findings of this study are available within the article and its Supplementary Information files or from the corresponding authors upon reasonable request. The Hi-C experimental data used in this study is available publicly, and corresponding SRA entries listed in Supplementary Data panel 1. Processed data is available from the authors upon request. A reporting summary for this Article is available as a Supplementary Information file.

## Code availability
Binless is an R/C++ package using gfl[37] and is available at https://github.com/3DGenomes/binless. We used Stan[33] (https://mc-stan.org) to prototype the statistical model.

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

## Acknowledgements
We are grateful to François Le Dily, Guillaume J. Filion, Francesca Di Giovanni, Simon Heath, Emanuele Raineri, and François Serra for fruitful discussion. Y.G.S would like to thank Theodore Sakellaropoulos and Aristotelis Tsirigos for their help in running the modified HiCRep lasso calculations in HiC-bench. This work has been partially

supported by the European Research Council under the European Union's Seventh Framework Programme (FP7/2007-2013)/ERC Synergy grant agreement 609989 (4Dgenome), the European Union's Horizon 2020 research and innovation programme (agreement 676556) as well as the Spanish MINECO (BFU2017-85926-P). We acknowledge support of the Spanish Ministry of Economy, Industry and Competitiveness (MEIC) to the EMBL partnership, the Centro de Excelencia Severo Ochoa and the CERCA Programme / Generalitat de Catalunya. We also acknowledge support of the Spanish Ministry of Economy, Industry and Competitiveness (MEIC) through the Instituto de Salud Carlos III, the Generalitat de Catalunya through Departament de Salut and Departament d'Empresa i Coneixement and the Co-financing by the Spanish Ministry of Economy, Industry and Competitiveness (MEIC) with funds from the European Regional Development Fund (ERDF) corresponding to the 2014-2020 Smart Growth Operating Program.

## Author contributions

Y.G.S., D.C., and M.A.M-R. designed the method. Y.G.S. and D.C. developed the method and implemented the package. Y.G.S., D.C., and E.V. processed the Hi-C datasets. Y.G.S. analyzed the datasets. Y.G.S., D.C., and M.A.M-R. wrote the manuscript.

## Additional information

**Competing interests:** The authors declare no competing interests.

