## [Peer Review File · Nature Communications]

Reviewers' comments:

Reviewer #1 (Remarks to the Author):

In Spill et al., the authors propose a method for detecting interactions in Hi-C data. The proposed method is based on:

- A new classification of the paired-end reads obtained from Hi-C
- A resolution-free normalization
- Detection of interactions using fused lasso

Overall this work seems to be at a rather preliminary stage. The ideas are solid and there is an implemented method, however there is absolutely no comprehensive quantitative comparison to other methods in a large set of published Hi-C data. This is admittedly a lot of work, but absolutely necessary to allow the reviewers and future readers of this work to assess the performance of the proposed approach. Without such a benchmark, it is impossible to offer constructive feedback.

Examples of studies along these lines are reference #34 in the text, HiCRep (Yang et al., Genome Research 2017), and many others, including a preprint that uses fused lasso (<https://www.biorxiv.org/content/early/2017/11/01/141481>).

In the absence of a full benchmark of this method, I will provide a few major comments to improve this work:

- Provide comprehensive (many datasets) and quantitative (several metrics) evidence for all the claims made in this study
- Compare to other normalization approaches
- Compare to other methods that call specific interactions
- Benchmark the method at multiple sequencing depths (by resampling existing data) and at several resolutions
- Discuss reference #53 in more detail
- In general, provide evidence for all claims, e.g. "Two types of reads can rapidly be distinguished": examples and evidence needs to be provided.
- Page 5, "Second, binless also makes use of read pairs that are usually discarded, ...": provide quantitative evidence that this leads to better detection of interactions
- 4-cutters can indeed generate sparse matrices at low sequencing depth, but 6-cutters have a different type of sparsity (the maximum achievable resolution is around 5kb); please discuss this as well
- Does the proposed normalization correct for GC content and effective length? In other words, if the matrices are binned after the proposed normalization, do you see any biases?
- The section "Binned detection lacks sensitivity" needs to be heavily expanded to include convincing, comprehensive and quantitative comparisons to existing methods
- The same is true for the next section

Reviewer #2 (Remarks to the Author):

This manuscript presents a new algorithm named Binless that combines a novel strategy for Hi-C paired end read classification; a normalization and detection method; and a binless annotation strategy of Hi-C data.

The method could certainly have an interest in the field of chromosome conformation, and seems in principle to have the potential to overcome current limitations in Hi-C analysis pipelines. Its major quality seems to be the Binless detection part, which uses adaptive-resolution matrices that could really improve the way off-diagonal features are detected in Hi-C matrices, and revolutionise how Hi-C experiments are compared (which is notoriously difficult as soon as library complexity and/or sequencing depth differ even slightly across samples).

Unfortunately however the manuscript is very badly written, which makes it impossible to assess precisely how the algorithm works. The performances of the method are not benchmarked against any published alternative algorithm, the data presented are not convincing, and finally there is a number of non-explained (or incorrect) assumptions that require the analysis to be fundamentally improved and the manuscript to be entirely re-written to be understood (and make a real impact in the field).

Major points:

1. It is unclear how robust (or useful) the first two diagnostics related to sonication and DNA degradation, which are based on the proposed novel classification of reads, could actually be. The authors only show examples of 'good' Hi-C libraries in Figure 2A-B; it would be useful to show the same graphs for poorer quality libraries in order to give a sense of the effects that can be observed/distinguished, and how they can be related to library quality.

2. I have similar doubts on the third diagnostics (relative to the "ligation ratio"). First, also with reference at Figure 1B: what is the difference between 'dangling' and 'random' reads? I would have thought that what is called 'random' here is typically called 'dangling' in other algorithms used to process Hi-C reads (see for example Ref. 29), but I must have misunderstood? Second, if it correlates well with the percentage of cis interactions, one would be tempted to use the latter as a measure of ligation efficiency as it is currently done. I am not sure I understand the interest of the new annotation of reads at this point.

3. It is unfortunately very, very difficult to understand how the simultaneous bias removal and signal detection work in the Binless method (and to be honest how the entire package works!). The authors should consider rewriting the "Binned normalization" section entirely, so that it contains factual elements that would help the reader understand what exactly their algorithm does. In the current wording, it is not even clear what "signal detection" means to a non-Hi-C specialist (and still). In addition, the example provided in Figure 3 does not help understanding what the algorithm does. Does this synthetic map behave like an actual Hi-C map in terms of polymer scaling? If not, how can this be used to benchmark the performances of an algorithm that is designed to analyze data that live in a power-law scaling scenario? I had a hard time understanding what the results shown in Figure 3 mean, and why one should consider the 'Binless plus signal detection' results better than the others. I kind of see what this points at - but if the aim is to show that this last variant of the algorithm is able to pick up the off-diagonal interaction without making any assumptions on equal coverage, this should be definitely better explained and substantiated with more realistic examples.

4. The Hi-C examples shown in Figure 4-6 are not easy to interpret because of the choice of the color code. Also, it would be beneficial to enlarge the plotted region to emphasize that we are looking at a single TAD. Finally, chromosome numbers are missing and it is not clear what 'TBX3 locus' means: shall I interpret the locus of the TBX3 gene in human cells?

5. Figure 4D (referenced in the text) is missing!

6. What does 'statistical significance' mean in the context of a Fused Lasso regression?

7. More generally, and on the same line as point 3, the 'Binless detection' section does not explain exactly what the algorithm does - it rather seems more concerned with boasting the merits of it. For example, one key (and very clever!) idea here seems to be that binless matrices might be a much better way to plot Hi-C data and assess robust features that stand out against the random polymer behavior. But how are these matrices built? Do they emerge as a consequence of the Fused Lasso regression or are they an independent feature implemented by the code? This is absolutely incomprehensible in the text and would deserve a much clearer explanation.

8. How are spurious peaks defined, and based on which criteria they are removed by Binless?

9. A measure of the "dramatic increase in detection sensitivity" by Binless is missing: please provide a benchmark for the performance of the code.

10. How are "significantly enriched" loops defined? I couldn't find any description of which criteria are used. Also, and importantly: Binless is claimed to be a much better alternative to current loop caller algorithms, but it seems here that some additional criterion (external to Binless) is needed to identify the loops in the binless matrices. If not, then the authors should absolutely clarify this point. Otherwise how can "six loops" be defined in Fig. 7?

11. Figures 6A and B are not referenced in the text!

12. How are "significant differences" between replicates defined?

13. How are "false discoveries" defined in Table 1? With respect to which criteria?

Minor points:

1. Introduction: Genes within TADs do not tend to be 'co-expressed' but to be coregulated during cell differentiation of external stimulation.

2. It is not clear in the introduction why loop calling is in general problematic.

3. What are counter-diagonal biases in raw Hi-C matrices?

4. Figure 1B: arrows are too small and colors too similar to really understand what is plotted near the diagonal. What is the difference between 'other' and 'random' reads?

5. Figure 4A-B: what is the bin size there? Which threshold is chosen to call significant interactions?

Reviewer #3 (Remarks to the Author):

Spill and collaborators introduce a new method to represent, normalize and detect chromatin conformation interactions that does not require binning of the genome. Thus, compared to available methods the so called 'binless' method has the advantage of being resolution free. The authors claim that their method can better normalize the inherent bias of Hi-C data. Also, the authors provide a method to detect Hi-C features like loops or TADs that is also resolution free.

In my opinion the paper explores a novel ways to treat Hi-C data that I think is useful for the researchers working on the field, specially for cases in which the Hi-C reads have been enriched in some way. However, in my opinion the manuscript requires some further work to present their method and their results more clearly. Following are some general suggestions to improve the manuscript and some questions.

1. The proposed normalization seems impractical for almost all real use cases as it is only limited to 100 restriction sites (even bacterial genomes have thousands restriction sites). For which applications is this normalization useful?

2. The fast approximation aims to improve over the limitations of the exact method. But, does this approximation also has limitations? Can you provide some information on the time and memory required to run Binless on some genomes (eg. human, mouse, worm, fly or yeast genome)? I presume that the results presented on the paper used this approximation. You should make this clear since the beginning.

3. The first paragraph of the discussion offers a more clear summary and justification for the paper than the introduction. I would suggest to move it to the beginning. Contrary to the introduction and results section, the discussion is more clear about what I think is the main message of the paper: a novel normalization method and a novel feature detection (fused lasso).

4. The first section of the results (Base-resolution view of hi-C data) is distracting with respect to the main message. Although I like how the authors represent the Hi-C data using arrows, the use of this representation is orthogonal to the main message and not used afterwards. What is important is the classification of pairs that is then used for the normalization. My suggestion is reduce this part and focus on the next section. Put the figures as supplement and use Figure 1 to explain the normalization method and the Hi-C pair classification strategy. Similarly, Figure 2 seems like standard quality control and does not merit to be a main Figure.

5. Since this is a methods paper, the section (Binless normalization) should contain more details to understand the justification for the method. Most of the current methods section should be part of this section. Hopefully, the authors can add a visual description of their method to help to understand it's merits. I found Supplementary Figure 3 quite relevant to the method, but is not a main Figure.

6. The authors said that they build upon the HiCNorm negative binomial regression framework. However, since the paper relies heavily in the use of negative binomials I find important to offer a clear justification for the use of this distribution.

7. Something I did not understand is why the authors have a full section called Binned detection (with Figure 4), to then argue that binned detection lacks sensitivity and should be replaced by their Binless detection. Thus, I think this section needs to be modified or removed and instead highlight and add more details their binless detection.

Other points

1. In the results section, the authors define the LR ratio, however this definition seems different than the LR ratio definition in methods. Furthermore, the authors use 'reads', in reference to their representation of Hi-C data pairs. But since 'reads' is normally used in the context of NGS sequencing data, frases like 'reads close to the diagonal' are not obvious to understand. I suggest that the authors use some other name like 'Hi-C pairs'.

2. The reason for the classification of Hi-C pairs is not entirely clear to me. There is some sort of standard classification of reads that can be seen for example in Figure 4 of HiCPro publication in Genome Biology but is also present in the quality control of many Hi-C publications. So read classification is nothing new. The class 'contact far', following the example on Figure 1D, contains inward reads that are separated by more than one restriction site (I assume as this is not clear). They could be separated by few bp or millions of bp. So the 'far' description is relative. Similarly, the class 'contact close' seems to contain outward facing reads (Fig. 1D). If they are flanked by two neighboring restriction sites, they are usually classified as 'self-circles', but not all outward facing reads are self circles or contain reads that are linearly close. I presume that 'self-circles' have their own category based on Fig. 1B. Finally, if the both reads map on the forward strand they are classified as 'contact up' and if both map on the reverse strand they are classified as 'contact down'. In general, why not call these pairs: 'inward', 'outward', 'both reverse', 'both forward' or something along this lines that make it clear what they are. Also, in Figure 1, the D part should be first as this explains the classification, then C which shows how the arrows are placed, then A and B.

In the supplementary methods, negative binomials are part of the model for c_{far} , c_{close} , c_{up} , c_{down} . However, I don't understand why those classes need to be treated differently. Hi-C pairs separated some kb can be in any orientation with respect to each other given the stochasticity of

the ligation events. Maybe the authors can justify better their classification.

3. In the supplement, the description of the 'Exact Model' doesn't explain what RJ, DL and DR are. The authors should make an effort to explain their model as this is the core of the manuscript.

4. In the binless normalization section, the authors write: "at constant digestion rate, the number of dangling reads would drop with increase efficiency of ligation". As far as I know, the digestion rate applies to restriction enzymes, but I think that what the authors mean is constant ligation rate. Please revise.

5. In the same section. "First, normalization is performed prior to binning the data". This means that the normalization is applied per read or Hi-C pair? Or is this done by restriction site or restriction fragment?

6. The 'equal visibility assumption' is not used by the authors but this is not well justified. I could think that in methods like Hi-C chip or capture-C where certain regions are enriched the 'equal visibility assumption' does not hold but otherwise, this assumption is well justified. The use of the fake data to justify the invalidity of the equal visibility assumption does not seem to be fair because the construction of the fake data can be adjusted to suit the argument.

7. In the section 'Binned detection' the authors say "the polymer effect must be accounted for". For clarity, the authors should explain that they refer to the exponential decay of contacts with genomic distance. 'Polymer effect' could refer to a number of other unrelated things.

8. In the binless detection section says: "a binless matrix is a matrix whose bins adapt to the size of the features detected". I find confusing that a binless matrix has bins. Also, the matrix representation used by the authors is not described. The binless matrix is what exactly? The Fused Lasso regression is applied to this matrix?

9. The methods section says that the input for the binless software is the reads intersection file of TADbit. But in the 'Figure and table generation' says that the input for binless are mapped pair-end reads (bam files presumably). Please revise which is correct.

10. In methods says: "Cis-trans ratio: Number of filtered reads arising within chr1, divided by total number of filtered reads with one end mapping to chr1". Is it 'with one end mapping to chr1 and the other end mapping to other chromosome' ? En general, for clarity I prefer to use intra-chromosomal and inter-chromosomal contacts to avoid confusion.

Reviewer #1 (Remarks to the Author):

In Spill et al., the authors propose a method for detecting interactions in Hi-C data. The proposed method is based on:

- *A new classification of the paired-end reads obtained from Hi-C*
- *A resolution-free normalization*
- *Detection of interactions using fused lasso*

Overall this work seems to be at a rather preliminary stage. The ideas are solid and there is an implemented method, however there is absolutely no comprehensive quantitative comparison to other methods in a large set of published Hi-C data. This is admittedly a lot of work, but absolutely necessary to allow the reviewers and future readers of this work to assess the performance of the proposed approach. Without such a benchmark, it is impossible to offer constructive feedback. Examples of studies along these lines are reference #34 in the text, HiCRep (Yang et al., Genome Research 2017), and many others, including a preprint that uses fused lasso

(<https://www.biorxiv.org/content/early/2017/11/01/141481>).

In the absence of a full benchmark of this method, I will provide a few major comments to improve this work:

- *Provide comprehensive (many datasets) and quantitative (several metrics) evidence for all the claims made in this study*
- *Compare to other normalization approaches*
- *Compare to other methods that call specific interactions*
- *Benchmark the method at multiple sequencing depths (by resampling existing data) and at several resolutions*
- *Discuss reference #53 in more detail*
- *The section "Binned detection lacks sensitivity" needs to be heavily expanded to include convincing, comprehensive and quantitative comparisons to existing methods*
- *The same is true for the next section*

We performed a large-scale benchmark by examining the datasets studied in the milestone comparison by Forcato *et al.* in Nature Methods (ref #34). We subjected these datasets to binless normalization, and compared it to other methods, including ICE, shaman (ref #53), HiCRep and diffHiC. The new benchmark is described in page 5 and the results are summarized in Figures 2 to 5 and Sup. Fig. 6 to 9. We compared reproducibility of normalized matrices using several metrics. We also examined interaction and difference detection capabilities using experimentally validated interactions, and numerous side-by-side examples. We believe we proved that Binless is on par and often even outperforms other methods on all these aspects.

- *In general, provide evidence for all claims, e.g. "Two types of reads can rapidly be distinguished": examples and evidence needs to be provided.*
- *Page 5, "Second, binless also makes use of read pairs that are usually discarded, ...": provide quantitative evidence that this leads to better detection of interactions*

We removed these comments since they did not add much value to the manuscript. Use of reads aligning close to the diagonal is a necessity in binless, and cannot be avoided. Indeed, for observed counts, the genomic bias contribution is in the form $b_i b_j$ and remains constant when transforming $b \rightarrow -b$. This degeneracy is removed by the likelihoods for dangling and rejoined ends, which only contain one such biasing term and not a product.

- *4-cutters can indeed generate sparse matrices at low sequencing depth, but 6-cutters have a different type of sparsity (the maximum achievable resolution is around 5kb); please discuss this as well*

We expanded the section in the discussion (page 5, 2nd paragraph) and in the methods section (page 11) in which we recommend sensible values for the base resolution.

- Does the proposed normalization correct for GC content and effective length? In other words, if the matrices are binned after the proposed normalization, do you see any biases?

Binless corrects for any genomic biases without specifying them. It behaves like iterative correction. After normalization, we do not see any biases (Sup. Fig. 15, “normalized” matrix) but GC content and effective length were not used to produce that result.

Reviewer #2 (Remarks to the Author):

This manuscript presents a new algorithm named Binless that combines a novel strategy for Hi-C paired end read classification; a normalization and detection method; and a binless annotation strategy of Hi-C data.

The method could certainly have an interest in the field of chromosome conformation, and seems in principle to have the potential to overcome current limitations in Hi-C analysis pipelines. Its major quality seems to be the Binless detection part, which uses adaptive-resolution matrices that could really improve the way off-diagonal features are detected in Hi-C matrices, and revolutionise how Hi-C experiments are compared (which is notoriously difficult as soon as library complexity and/or sequencing depth differ even slightly across samples).

Unfortunately however the manuscript is very badly written, which makes it impossible to assess precisely how the algorithm works. The performances of the method are not benchmarked against any published alternative algorithm, the data presented are not convincing, and finally there is a number of non-explained (or incorrect) assumptions that require the analysis to be fundamentally improved and the manuscript to be entirely re-written to be understood (and make a real impact in the field).

We rewrote the manuscript, included a large benchmark (page 5, Fig. 2 to 5 and Sup. Fig. 6 to 9.), and left out contributions that were distracting with respect to the main message. As such, we removed the “binned detection” part, former figure 4, and moved former Fig. 2 to the supplementaries, modifying it to keep only diagnostics relevant for normalization.

Major points:

1. It is unclear how robust (or useful) the first two diagnostics related to sonication and DNA degradation, which are based on the proposed novel classification of reads, could actually be. The authors only show examples of 'good' Hi-C libraries in Figure 2A-B; it would be useful to show the same graphs for poorer quality libraries in order to give a sense of the effects that can be observed/distinguished, and how they can be related to library quality.

We modified this figure (now Sup. Fig. 11) by showing two datasets per plot, to illustrate the variability between experimental conditions. We chose them to represent the diversity found in the benchmark datasets. These plots serve mostly to determine parameters for Binless. For the experimentalist, we don't believe these plots are immediately useful, since they will have done most of the checks before sequencing the library.

2. I have similar doubts on the third diagnostics (relative to the "ligation ratio"). First, also with reference at Figure 1B: what is the difference between 'dangling' and 'random' reads? I would have thought that what is called 'random' here is typically called 'dangling' in other algorithms used to process Hi-C reads (see for example Ref. 29), but I must have misunderstood? Second, if it correlates well with the percentage of cis interactions, one would be tempted to use the latter as a measure of ligation efficiency as it is currently done. I am not sure I understand the interest of the new annotation of reads at this point.

We removed the third diagnostic, which did not add much to the paper and is not required to run Binless, contrary to the two other diagnostics (Sup. Fig. 11). As for dangling ends, our

definition is stricter. Indeed, in former ref 29, dangling ends refer to what would be dangling + random reads.

3. It is unfortunately very, very difficult to understand how the simultaneous bias removal and signal detection work in the Binless method (and to be honest how the entire package works!). The authors should consider rewriting the "Binned normalization" section entirely, so that it contains factual elements that would help the reader understand what exactly their algorithm does. In the current wording, it is not even clear what "signal detection" means to a non-Hi-C specialist (and still). In addition, the example provided in Figure 3 does not help understanding what the algorithm does. Does this synthetic map behave like an actual Hi-C map in terms of polymer scaling? If not, how can this be used to benchmark the performances of an algorithm that is designed to analyze data that live in a power-law scaling scenario? I had a hard time understanding what the results shown in Figure 3 mean, and why one should consider the 'Binless plus signal detection' results better than the others. I kind of see what this points at - but if the aim is to show that this last variant of the algorithm is able to pick up the off-diagonal interaction without making any assumptions on equal coverage, this should be definitely better explained and substantiated with more realistic examples.

We hope to have addressed this major issue. We added several flowcharts explaining in detail how the fitting is performed (Sup. Fig. 1b) and how the data is pre- and post-processed (Sup. Fig. 1a,c). We reorganized the whole paper and used as a starting point the new figure 1, which synthesizes what is happening in Binless. We left out the synthetic benchmark as well as the intermediate "binned detection" section, as they were superseded by the benchmark we present now and distracted from the main message.

4. The Hi-C examples shown in Figure 4-6 are not easy to interpret because of the choice of the color code. Also, it would be beneficial to enlarge the plotted region to emphasize that we are looking at a single TAD. Finally, chromosome numbers are missing and it is not clear what 'TBX3 locus' means: shall I interpret the locus of the TBX3 gene in human cells?

We enlarged the regions in all the plots we now show (Fig. 5 and Sup. Fig. 8 and 9), and specified the locations of the considered loci more clearly (apart from Fig. 1 which is meant as an illustration).

5. Figure 4D (referenced in the text) is missing!

It was meant to be Sup. Fig. 4D. In any case, we removed this part.

6. What does 'statistical significance' mean in the context of a Fused Lasso regression?

The sparse fused lasso regression proposed here is laid out in a Bayesian framework. Statistically significant signal means that it is more probable that there is a nonzero signal in that region than not.

7. More generally, and on the same line as point 3, the 'Binless detection' section does not explain exactly what the algorithm does - it rather seems more concerned with boasting the merits of it. For example, one key (and very clever!) idea here seems to be that binless matrices might be a much better way to plot Hi-C data and assess robust features that stand out against the random polymer behavior. But how are these matrices built? Do they emerge as a consequence of the Fused Lasso regression or are they an independent feature implemented by the code? This is absolutely incomprehensible in the text and would deserve a much clearer explanation.

We added several elements across the paper to highlight the construction and interpretation of binless signal and difference matrices. The beginning of the results section (page 5) summarizes the operations in Binless and points to Fig. 1, a graphical summary. Sup. Fig. 1 gives detailed flowcharts on the procedure. The discussion (page 7, 2nd and 3rd paragraph) explains how binless signal and difference matrices are estimated. We hope to have clarified this point.

8. How are spurious peaks defined, and based on which criteria they are removed by Binless?

We expanded the discussion (page 7, 2nd and 3rd paragraph) to cover this point.

9. A measure of the "dramatic increase in detection sensitivity" by Binless is missing: please provide a benchmark for the performance of the code.

We removed the dramatic tone in our writing and provided the large benchmark requested (page 5, Fig. 2 to 5 and Sup. Fig. 6 to 9.) which, we hope, proves our point more quantitatively.

10. How are "significantly enriched" loops defined? I couldn't find any description of which criteria are used. Also, and importantly: Binless is claimed to be a much better alternative to current loop caller algorithms, but it seems here that some additional criterion (external to Binless) is needed to identify the loops in the binless matrices. If not, then the authors should absolutely clarify this point. Otherwise how can "six loops" be defined in Fig. 7?

We conclude our discussion with this point (page 7, last paragraph)

11. Figures 6A and B are not referenced in the text!

We removed this figure

12. How are "significant differences" between replicates defined?

We added comments in that respect in the methods section (page 13, last paragraph)

13. How are "false discoveries" defined in Table 1? With respect to which criteria?

Table 1 has been replaced, and this section superseded by the benchmark.

Minor points:

1. Introduction: Genes within TADs do not tend to be 'co-expressed' but to be coregulated during cell differentiation of external stimulation.

We fixed this sentence.

2. It is not clear in the introduction why loop calling is in general problematic.

We clarified the sentence pointing at poor replicability of interactions between replicates.

3. What are counter-diagonal biases in raw Hi-C matrices?

These refer to the strong decay in interaction probability with the distance between loci.

4. Figure 1B: arrows are too small and colors too similar to really understand what is plotted near the diagonal. What is the difference between 'other' and 'random' reads?

Random reads are reads which do not cross a restriction site, and whose ends map to the same fragment. Other reads are those that could not be classified, such as the one that can be seen in the middle of the “contact up” reads.

5. Figure 4A-B: what is the bin size there? Which threshold is chosen to call significant interactions?

This figure and binwise interaction detection were removed.

Reviewer #3 (Remarks to the Author):

Spill and collaborators introduce a new method to represent, normalize and detect chromatin conformation interactions that does not require binning of the genome. Thus, compared to available methods the so called 'binless' method has the advantage of being resolution free. The authors claim that their method can better normalize the inherent bias of Hi-C data. Also, the authors provide a method to detect Hi-C features like loops or TADs that is also resolution free.

In my opinion the paper explores a novel ways to treat Hi-C data that I think is useful for the researchers working on the field, specially for cases in which the Hi-C reads have been enriched in some way. However, in my opinion the manuscript requires some further work to present their method and their results more clearly. Following are some general suggestions to improve the manuscript and some questions.

1. The proposed normalization seems impractical for almost all real use cases as it is only limited to 100 restriction sites (even bacterial genomes have thousands restriction sites). For which applications is this normalization useful?

The text was confusing in that respect, and we hope to have clarified it (Methods section, page 10, first and following paragraphs). In brief, the proposed model has $4N^2$ negative binomial likelihoods for N cut sites, and their evaluation is cumbersome because they involve the gamma function, whose values are slow to evaluate. Instead, the IRLS approximation (former section "fast approximation" in the Sup. Mat.) was shown since the 1970s to converge to the exact solution, and leads to great speedups. It makes the exact model applicable to regions of up to about 4Mb.

2. The fast approximation aims to improve over the limitations of the exact method. But, does this approximation also has limitations? Can you provide some information on the time and memory required to run Binless on some genomes (eg. human, mouse, worm, fly or yeast genome)? I presume that the results presented on the paper used this approximation. You should make this clear since the beginning.

We clarified the presentation (methods section, page 10, 6th paragraph), and also added Sup. Fig. 12, which shows both CPU and memory consumption.

3. The first paragraph of the discussion offers a more clear summary and justification for the paper than the introduction. I would suggest to move it to the beginning. Contrary to the introduction and results section, the discussion is more clear about what I think is the main message of the paper: a novel normalization method and a novel feature detection (fused lasso).

We moved that section to the introduction.

4. The first section of the results (Base-resolution view of hi-C data) is distracting with respect to the main message. Although I like how the authors represent the Hi-C data using arrows, the use of this representation is orthogonal to the main message and not used afterwards. What is important is the classification of pairs that is then used for the normalization. My suggestion is reduce this part and focus on the next section. Put the figures as supplement and use Figure 1 to explain the normalization method and the Hi-C pair classification strategy. Similarly, Figure 2 seems like standard quality control and does not merit to be a main Figure.

We completely rewrote the paper, following the suggestion of the reviewer. We reorganized the presentation around binless interaction and difference detection. Arrow plots were moved to the end of the results. Other intermediate results, such as binned interaction detection, were discarded.

5. Since this is a methods paper, the section (Binless normalization) should contain more details to understand the justification for the method. Most of the current methods section should be part of this section. Hopefully, the authors can add a visual description of their method to help to understand it's merits. I found Supplementary Figure 3 quite relevant to the method, but is not a main Figure.

We made substantial changes to the figures. Figure 1 is now a visual overview of what Binless has to offer. We also added several flowcharts (Sup. Fig. 1 and 10) to explain its inner workings.

6. The authors said that they build upon the HiCNorm negative binomial regression framework. However, since the paper relies heavily in the use of negative binomials I find important to offer a clear justification for the use of this distribution.

The use of the negative binomial is very widespread in high-throughput sequencing experiments. It is the gaussian of integers, meaning it offers much more flexibility than the Poisson distribution, whose variance is fixed with respect to the mean. Recent experiments in Syn-HiC (ref. 67) actually prove it is a pertinent choice for Hi-C data (see results, page 4, last paragraph).

7. Something I did not understand is why the authors have a full section called Binned detection (with Figure 4), to then argue that binned detection lacks sensitivity and should be replaced by their Binless detection. Thus, I think this section needs to be modified or removed and instead highlight and add more details their binless detection.

We removed that section since what it tried to prove is demonstrated through the benchmark.

Other points

1. In the results section, the authors define the LR ratio, however this definition seems different than the LR ratio definition in methods. Furthermore, the authors use 'reads', in reference to their representation of Hi-C data pairs. But since 'reads' is normally used in the context of NGS sequencing data, frases like 'reads close to the diagonal' are not obvious to understand. I suggest that the authors use some other name like 'Hi-C pairs'.

The LR was removed because it was distracting and of no use for binless normalization. We use "arrows" instead of reads to clarify the presentation (page 6, first paragraph).

2. The reason for the classification of Hi-C pairs is not entirely clear to me. There is some sort of standard classification of reads that can be seen for example in Figure 4 of HiCPro publication in Genome Biology but is also present in the quality control of many Hi-C publications. So read classification is nothing new. The class 'contact far', following the example on Figure 1D, contains inward reads that are separated by more than one restriction site (I assume as this is not clear). They could be separated by few bp or millions of bp. So the 'far' description is relative. Similarly, the class 'contact close' seems to contain outward facing reads (Fig. 1D). If they are flanked by two neighboring restriction sites, they are usually classified as 'self-circles', but not all outward facing reads are self circles or contain reads that are linearly close. I presume that 'self-circles' have their own category based on Fig. 1B. Finally, if the both reads map on the forward strand they are classified as 'contact up' and if both map on the reverse strand they are classified as 'contact down'. In

general, why not call these pairs: 'inward', 'outward', 'both reverse', 'both forward' or something along this lines that make it clear what they are. Also, in Figure 1, the D part should be first as this explains the classification, then C which shows how the arrows are placed, then A an B.

We reorganized Fig. 6 (former Fig. 1) as suggested. However, we keep the nomenclature of the contacts, but dwell more on the explanation (page 6, first paragraph). Up contacts are upstream of the restriction site, etc.

In the supplementary methods, negative binomials are part of the model for c_{far} , c_{close} , c_{up} , c_{down} . However, I don't understand why those clases need to be treated differently. Hi-C pairs separated some kb can be in any orientation with respect to each other given the stochasticity of the ligation events. Maybe the authors can justify better their classification.

We agree ligation events are stochastic, but the ligase is affected by sequence-specific traits such as local GC content or DNA stiffness. Since these are sequence-specific, we do not expect them to be exactly equivalent on either side of a cut site. We therefore chose to design our regression by splitting each contact in four categories.

3. In the supplement, the description of the 'Exact Model' doesn't explain what RJ, DL and DR are. The authors should make an effort to explain their model as this is the core of the manuscript.

We improved the presentation of the supplementary material (section 2.1 page 2).

4. In the binless normalization section, the authors write: "at constant digestion rate, the number of dangling reads would drop with increase efficiency of ligation". As far as I know, the digestion rate applies to restriction enzymes, but I think that what the authors mean is constant ligation rate. Please revise.

We rephrased the sentence (page 6 3rd paragraph). When one cut site intersection has worse ligation efficiency than another, we expect an enrichment in dangling reads at that cut site. But this is true only if the efficiency of *digestion* is the same in both cases, otherwise they cannot be compared.

5. In the same section. "First, normalization is performed prior to binning the data". This means that the normalization is applied per read or Hi-C pair? Or is this done by restriction site or restriction fragment?

We clarified this point (page 6 3rd paragraph). Binless normalization indeed happens at cut-site level.

6. The 'equal visibility assumption' is not used by the authors but this is not well justified. I could think that in methods like Hi-C chip or capture-C where certain regions are enriched the 'equal visibility assumption' does not hold but otherwise, this assumption is well justified. The use of the fake data to justify the invalidity of the equal visibility assumption does not seem to be fair because the construction of the fake data can be adjusted to suit the argument.

The fake data was superseded by the benchmark, so we removed it. We agree with the reviewer that fake data has limitations, and we did not find it useful to enter this discussion in the manuscript given the abundance of experimental data. All regression-based normalization methods (among which diffHiC and oneD) relax the equal visibility assumption, and allow small deviations around an average value.

7. In the section 'Binned detection' the authors say "the polymer effect must be accounted for". For clarity, the authors should explain that they refer to the exponential decay of contacts with genomic distance. 'Polymer effect' could refer to a number of other unrelated things.

We rephrased these and similar lines (page 9 last paragraph).

8. In the binless detection section says: "a binless matrix is a matrix whose bins adapt to the size of the features detected". I find confusing that a binless matrix has bins. Also, the matrix representation used by the authors is not described. The binless matrix is what exactly? The Fused Lasso regression is applied to this matrix?

We hope to have clarified this important point. The binless matrix pixels can be seen as the pixels of a detector. Neighboring pixels were fused to have the same value when the underlying Hi-C data was estimated to have a common signal. It can therefore be seen as a collection of patches of varying sizes and shapes (see results, page 4, 3rd paragraph and discussion, page 7, 1st paragraph).

9. The methods section says that the input for the binless software is the reads intersection file of TADbit. But in the 'Figure and table generation' says that the input for binless are mapped pair-end reads (bam files presumably). Please revise which is correct.

We fixed this, and also provided a flowchart (Sup. Fig. 1a). The input is a TADbit tsv file.

10. In methods says: "Cis-trans ratio: Number of filtered reads arising within chr1, divided by total number of filtered reads with one end mapping to chr1". Is it 'with one end mapping to chr1 and the other end mapping to other chromosome' ? En general, for clarity I prefer to use intra-chromosomal and inter-chromosomal contacts to avoid confusion.

We removed this part.

REVIEWERS' COMMENTS:

Reviewer #1 (Remarks to the Author):

The authors have conducted an extensive benchmark of their methods and I believe they have convincingly showed that their method is at least as good as the best of the published methods. In some metrics, it is actually demonstrated to be superior (with one caveat mentioned below).

I have a couple of remaining comments:

Figure 2: "each comparison is made on a whole chromosome", please clarify this statement; I am assuming that this means that you computed SCCs on each chromosome and the boxplots show the distribution of SCC values across chromosomes, but please clarify for the benefit of the reader.

Fused lasso: I think the authors are confused about which paper proposed fused 2D lasso for Hi-C. It is not HiCRep, but this one instead: Gong et al., Nat Comm, <https://www.nature.com/articles/s41467-018-03017-1>. Please cite it in the introduction.

Also, based on this comment, I am not sure whether they actually compared to HiCRep or Gong et al. (HiC-bench version 2). HiCRep uses a different approach, mean 2D filter. Can you clarify? The correct comparison should be with Gong et al., although ideally you want to have both HiC-bench and HiCRep.

One final comment: In the case of detecting specific interactions (as opposed to TADs), Gong et al. propose a distance-normalization approach before applying fused 2D lasso. Was this done? If not, then I do expect to see good reproducibility (as demonstrated in Fig 2), but at the same time low sensitivity in detecting loops (I believe that is demonstrated in Fig 3). I understand that the authors have already put a lot of effort in revising their manuscript, so it may be too much to redo everything. Instead, they can make appropriate comments in the text.

Reviewer #2 (Remarks to the Author):

The authors have addressed most of the issues that I had previously raised, and notably provided a benchmark against several other methods for normalization of Hi-C data. Regrettably however, the manuscript remains by and large unintelligible. This is not a secondary point. I am really convinced that Binless is a great method, which will potentially become widely used by the growing community of researchers using Hi-C. However, very much like in the first submission (or maybe even worse), the text is really (really!) obscure. Even in the new version, there is absolutely no explanation (at least in the main text) of what *exactly* Binless is based on, other than some very abstract references to fused Lasso and Generalized Additive Model. This is a very innovative and potentially transformative method and the readers must be allowed to understand what it is about: otherwise no one will ever use it! For example, Figure S1 is really impossible to understand without some kind of explanation in the main text. In fact the text is so obscure that it is not even clear what Binless actually DOES: does it only normalize the data, or does it provide calls for where TAD borders or loops actually are? I could provide a long list of suggestions on how to make the manuscript clearer, but I do not think that this falls under this Reviewer's responsibility. I would like to wholeheartedly encourage the Authors to re-think and re-write the manuscript in an understandable format that matches the standards of manuscripts where new methods are presented.

Finally, on a more technical side, I would like to point out that it is not clear what the SCC and 'tpr'

metrics are, and that it would be very beneficial not only to better explain them to the reader but also to provide examples of what some of the least-performing algorithms predict vs. the Binless prediction.

Reviewer #1 (Remarks to the Author):

The authors have conducted an extensive benchmark of their methods and I believe they have convincingly showed that their method is at least as good as the best of the published methods. In some metrics, it is actually demonstrated to be superior (with one caveat mentioned below).

I have a couple of remaining comments:

Figure 2: "each comparison is made on a whole chromosome", please clarify this statement; I am assuming that this means that you computed SCCs on each chromosome and the boxplots show the distribution of SCC values across chromosomes, but please clarify for the benefit of the reader.

We hope the legend is clearer now

Fused lasso: I think the authors are confused about which paper proposed fused 2D lasso for Hi-C. It is not HiCRep, but this one instead: Gong et al., Nat Comm, <https://www.nature.com/articles/s41467-018-03017-1>. Please cite it in the introduction.

Also, based on this comment, I am not sure whether they actually compared to HiCRep or Gong et al. (HiC-bench version 2). HiCRep uses a different approach, mean 2D filter. Can you clarify? The correct comparison should be with Gong et al., although ideally you want to have both HiC-bench and HiCRep.

We thank the reviewer for spotting this inconsistency. In fact, what we call HiCRep throughout the paper is a modification of the HiCRep protocol (by HiC-bench authors) which uses fused lasso instead of 2D mean filtering. The suggestion to still call it HiCRep comes from discussions with Theodore Sakellaropoulos (and Aristotelis Tsirigos), authors of the Gong et al. paper. They point out that HiCRep is the normalization procedure (which they modify slightly), while HiC-bench is the whole data processing pipeline. We clarified the manuscript in that regard.

One final comment: In the case of detecting specific interactions (as opposed to TADs), Gong et al. propose a distance-normalization approach before applying fused 2D lasso. Was this done? If not, then I do expect to see good reproducibility (as demonstrated in Fig 2), but at the same time low sensitivity in detecting loops (I believe that is demonstrated in Fig 3). I understand that the authors have already put a lot of effort in revising their manuscript, so it may be too much to redo everything. Instead, they can make appropriate comments in the text.

We included this distance-normalized matrix in several parts of the paper, and refer to it as HiCRep z-score matrix. As explained in the methods section, for all pairwise comparisons, we

use matrices with diagonal decay (that would be the standard HiCRep lasso-corrected matrix). For all interaction detections, we use matrices without diagonal decay (except for the raw data) (here, we would use the HiCRep lasso-corrected z-score matrix). The results are in line with the expectations of the reviewer: HiCRep is very reproducible (Figure 2), but the true positive rate remains comparable to the other methods. In the discussion, we argue that our using the weighted version of the fused lasso and a proper statistical modelling of these weights is what allows us to achieve better results.

Reviewer #2 (Remarks to the Author):

*The authors have addressed most of the issues that I had previously raised, and notably provided a benchmark against several other methods for normalization of Hi-C data. Regrettably however, the manuscript remains by and large unintelligible. This is not a secondary point. I am really convinced that Binless is a great method, which will potentially become widely used by the growing community of researchers using Hi-C. However, very much like in the first submission (or maybe even worse), the text is really (really!) obscure. Even in the new version, there is absolutely no explanation (at least in the main text) of what *exactly* Binless is based on, other than some very abstract references to fused Lasso and Generalized Additive Model. This is a very innovative and potentially transformative method and the readers must be allowed to understand what it is about: otherwise no one will ever use it! For example, Figure S1 is really impossible to understand without some kind of explanation in the main text. In fact the text is so obscure that it is not even clear what Binless actually DOES: does it only normalize the data, or does it provide calls for where TAD borders or loops actually are? I could provide a long list of suggestions on how to make the manuscript clearer, but I do not think that this falls under this Reviewer's responsibility.*

I would like to wholeheartedly encourage the Authors to re-think and re-write the manuscript in an understandable format that matches the standards of manuscripts where new methods are presented.

Finally, on a more technical side, I would like to point out that it is not clear what the SCC and 'tpr' metrics are, and that it would be very beneficial not only to better explain them to the reader but also to provide examples of what some of the least-performing algorithms predict vs. the Binless prediction.

We have rewritten the whole manuscript, and hope it is now much easier to read.